# Coping and Post-Traumatic Stress in Children and Adolescents after an Acute Onset Disaster: A Systematic Review

**DOI:** 10.3390/ijerph18094865

**Published:** 2021-05-03

**Authors:** Tara Powell, Kate M. Wegmann, Emily Backode

**Affiliations:** School of Social Work, University of Illinois, Urbana, IL 61801, USA; kwegmann@illinois.edu (K.M.W.); backode2@illinois.edu (E.B.)

**Keywords:** coping, post-traumatic stress, adolescents, children, disaster

## Abstract

Acute onset disasters impact children’s and adolescents’ psychological well-being, often leading to mental health challenges. The way a young person copes with the event plays a significant role in development of post-disaster psychopathology. Coping has been widely studied after acute onset disasters, however, difficulties conducting research in post-disaster contexts and the individualized nature of coping make accurate assessment of coping a significant challenge. A systematic literature search of multiple databases and previous reviews was conducted, exploring scholarly documentation of coping measurement and the relationship between coping and post-traumatic stress (PTSS) symptoms after acute onset disasters. A total of 384 peer-reviewed manuscripts were identified, and 18 articles met the inclusion criteria and were included in the current review. The studies examined coping and post-traumatic stress in the wake of acute onset disasters such as terrorist events and natural disasters, such as hurricanes, earthquakes, and wildfires. Greater PTSS symptoms were related to internalizing, externalizing, rumination, and avoidant coping strategies. Coping measurement was constrained due to measurement variance, lack of developmentally and contextually vali-dated instruments, theoretical misalignment, and absence of comprehensive tools to assess coping. Robust and consistent measures of coping should be established to inform research and interventions to reduce the impact of disasters on children’s and adolescents’ well-being.

## 1. Introduction

Sudden onset disasters, such as hurricanes, wildfires, tsunamis, and terrorist attacks leave survivors susceptible to adverse mental health outcomes, such as post-traumatic stress disorder (PTSD), depression, and anxiety [1,2,3]. Children and adolescents are among the most vulnerable to mental health difficulties after these disasters because they have less experience and knowledge on how to cope with these events [3,4]. In addition to disaster exposure, young people can experience a host of secondary stressors including separation from family and friends, loss of pets, displacement from home and school, and lack of basic needs, such as food, water, and medical care [4]. Common disaster-related distress reactions in children and adolescents include internalized symptoms, such as acute stress disorder, PTSD, depression, or generalized anxiety; and externalizing symptoms, such as behavior problems, substance abuse, and aggressive behaviors [5,6,7,8].

While many young people will experience disaster-related distress, most will adapt to their environment with only a subset experiencing sustained psychopathology. The development of psychological distress in children and adolescents has been closely linked to a myriad of risk and protective factors. Factors that put a young person at risk for mental health distress can include high levels of disaster exposure, low parental support, social isolation, poverty, and pre-existing mental health symptoms such as anxiety and depression [6,8,9]. Conversely, protective factors, such as peer social support, school connectedness, supportive parenting, problem solving, self-regulation skills, perceived self-efficacy, and positive maternal–child relationships can buffer the psychological impact of large-scale sudden-onset disasters [6,10].

The way children and adolescents cope may also serve as a protective or risk factor for post-disaster psychological symptoms. A young person’s ability to utilize adaptive or active coping methods, for example, can function as a barrier to the development of trauma-related symptomology [11]. Research has also indicated that acceptance, emotional expression, and cognitive reframing may buffer the psychological impact of a disaster on young people [6,12,13]. Alternatively, avoidant coping strategies, blame and anger, and social withdrawal have all been associated with higher levels of depression and post-traumatic stress symptoms in young people [14,15,16,17,18]. 

The accuracy of measures used to assess children’s coping and post-disaster mental health is essential in order to draw valid conclusions from study results. Post-disaster environments present unique challenges for conducting rigorous research [19]; one such challenge is finding valid, reliable psychological measures that can be feasibly accessed and administered within the constraints of a post-disaster setting. Balaban [20] recommended that post-disaster research should include relatively short, standardized questionnaires that can be administered by non-clinicians and that have previously demonstrated strong reliability and validity in post-disaster settings. Establishing strong psychometric properties for coping measures with the specific populations studied is necessary to ascertain content validity of the measure, given that coping is affected by a myriad of changing influences, such as individual characteristics, culture, and context. Even when examining psychometrically well-established measures, previous reviews have found a high degree of inconsistency in how dimensions of coping were described, making it difficult to draw valid conclusions across studies and measures [21]. The primary aim of this systematic review is to examine how child and youth coping is being measured following acute onset disasters, including: (1) what measures have been used, (2) dimensions of coping assessed, and (3) psychometric properties of the measures.

Because of the challenges in accurately assessing mental health constructs, such as post-disaster psychopathology and coping in young people, there is a lack of integration and consensus within the body of empirical evidence and research in this area. Therefore, the secondary aim, and substantive contribution, of this review is to examine and synthesize knowledge on the association between coping strategies and post-traumatic stress symptoms (PTS) after a disaster.

## 2. Method

### 2.1. Search Strategies

The Preferred Reporting Items for Systematic Reviews and Meta-Analyses (PRISMA) standards were used to search the EBSCOhost and Google Scholar electronic databases [22]. A total of four searches were completed, consisting of combinations of the following specific search terms: (disaster OR earthquake OR tsunami OR hurricane OR tornado OR typhoon OR flood OR avalanche OR landslide OR mudflow OR volcanic eruption OR eruption OR blizzards OR cyclonic storm OR cyclone OR hailstorm OR fire OR wildfire OR terrorist) and (Coping or Cope) and (Post-traumatic stress or PTSD or Post-traumatic stress disorder) and (Children or youth or youths or adolescents or adolescence or students). Searches were limited by the following specifications: full text availability, published in scholarly (peer reviewed) journals, and publication dates between the years of 1995 and the last date of the literature search (March 2021). Reference lists of related articles, such as the systematic review by Pfefferbaum et al. [21], were also examined as a check to ensure that the search had not inadvertently omitted potentially relevant studies.

### 2.2. Eligibility Criteria 

To be included, studies needed to assess coping as a primary variable of interest, along with measurement of PTSD using a psychometrically validated assessment aligned with DSM-III, DSM-IV, or DSM-V criteria. Eligible measures of “coping” were defined as those that assessed coping behaviors or beliefs in which child or adolescent participants directly engaged, such as coping strategies, coping efficacy beliefs, or coping assistance received from others. Because this review examined acute sudden onset mass trauma events, studies that examined chronic exposure to traumas (e.g., wars, on-going terror threats) were excluded.

Studies including children (6–12 years old) and adolescents (13–20 years old) were part of the inclusion criteria; excluded were those investigating preschool-aged (1–5 years old), infants (<1 year old), and emerging adults (>20 years old). Children younger than six were excluded because of the lack of developmentally appropriate self-report coping measures; emerging/young adults were excluded because they are of the developmental age to complete adult coping measures. 

Additional excluded publications were non-observational studies, studies that did not measure coping, studies not related to sudden onset disasters, studies not evaluating post-traumatic stress, review articles, studies evaluating therapeutic interventions, and qualitative studies. Included studies employed cross-sectional and longitudinal designs and used quantitative assessments of post-traumatic stress and coping. 

### 2.3. Articles Included

Our initial search string yielded 384 potentially appropriate publications. From these publications, 310 articles were removed after screening titles or abstracts. Full reports of 58 articles were reviewed. Forty articles were removed after full-text screening for the following reasons: thirty-two studies did not meet the criteria of measuring both PTSD and coping; three measured adult coping and PTSD; two were duplicates of the same study; three did not explore acute onset disasters. This yielded 18 articles included in the review. See Figure 1 for a PRISMA flow diagram. 

### 2.4. Data Extraction and Quality Appraisal

All studies were critically assessed and classified based on the type of disaster. Each study was extracted into an Excel spreadsheet. Quality assessments were conducted independently by two of the authors using the Systematic Assessment of Quality in Observational Research (SAQOR). The SAQOR is a tool developed by psychiatric researchers to assess the quality of observational studies [23]. The authors of the SAQOR note that it is especially intended to assess the quality of research on complex subjects, such as antidepressant use during pregnancy [23], psychosocial adjustment of former child soldiers [24], the association between depression and income inequality [25], and relationships between low self-esteem and internalizing disorders in youth [26], among other mental health topics. Given the applicability of the SAQOR to observational mental health research of a complex nature, the authors determined the SAQOR to be the most appropriate quality assessment for the studies included in the current review. The SAQOR evaluates key elements of rigor in observational studies: sample size and composition, quality of exposure and outcome measurements, follow-up (for longitudinal studies), distorting influences, reporting of data, and an overall assessment of study quality [23]. The authors of the current study slightly adapted the SAQOR to be more specific to the topic of post-disaster coping and PTSD, such that “exposure and outcome measurements” were defined as measures of coping and PTSD, respectively, and that “distorting influences” included a measure of disaster exposure as well as other potential confounders determined by each study’s research questions. Because control groups are typically not feasible or ethical in post-disaster research, the authors also eliminated the use of a control group as a measure of study quality. Out of the 18 reviewed articles, the authors disagreed on ratings for four studies. These disagreements were resolved through discussion until a mutual consensus was reached.

For each reviewed study, the type of disaster (flood, earthquake, wildfire, tsunami, hurricane, explosion, or terrorist attack), sample size, age of participants, country of participants, the time frame of the study in relation to the disaster, race/ethnicity of participants, and the coping and PTSD measures used were entered into a table (see Table 1). In addition, a brief summary of the findings related to the relationship between coping and PTSD as well as each study’s quality score also appear in Table 1. Features of the coping measures used in the reviewed studies were recorded in Table 2. Specific features of each measure included populations for which the measure has been validated, response style, dimensions of coping assessed, internal consistency scores, and effect sizes for the relationship between coping and PTSD. Effect sizes were recorded in the metric given in each article when such information was included. When a study did not include an effect size, one was calculated using data provided in the article whenever possible using Lenhard and Lenhard’s [27] effect size calculator. Although every effort was made to use a common metric of effect size, the variance in types of data provided meant that multiple metrics (e.g., Cohen’s *d*, Pearson’s *r*, odds ratio) were used to record effect sizes for as many studies as possible. 

## 3. Results

Of the eighteen studies included in the review, nine were cross-sectional and nine were longitudinal. Six articles were conducted after an earthquake, six after a hurricane, two after a wildfire, one in response to a flood, one after a tornado, one post-terrorist attack, and one that examined multiple disasters. These studies were conducted in five separate countries: United States (10 studies; 55.6%), China (5 studies; 27.8%), Italy (1 study; 5.5%), Greece (1 study, 5.5%), and Sri Lanka (1 study, 5.5%). Most of the manuscripts included elementary, middle, and high school aged children and youth; however, one included adolescents between the ages of 14–20 years. The age of the samples ranged from 7 to 20 years. The cross-sectional articles were assessed up to 8 years after the disaster, however, most were conducted within 2 years after the event. The longitudinal studies ranged between 10 days and 2 years post-disaster. Sample sizes ranged from small (46 children) to as large as 4118 children and youth. 

### 3.1. PTSD Measures

Nine measures of PTSD were used in the included studies. The most commonly used measures included the Child PTSD Symptom Scale (*n* = 7) [54] and the PTSD Reaction Index (*n* = 5) [55]. Other measures included the PTSD Checklist-Civilian Version [56], Children Revised Impact of Event Scale-13 (CRIES-13) [57], the PTSD Checklist [58]; the UCLA PTSD Index for DSM-IV [59], Impact of Events Scale [60], the PTSD Self-Rating scale [61], and the Trauma and Loss Spectrum (TALS) [62]. All included studies obtained data through self-report, with two studies also obtaining data through parent/caregiver report. Eight of the measures were based on the DSM-IV criteria for assessing PTSD and one measure (PTSD-RI) used the DSM-III-R criteria. Most studies (*n* = 17) estimated the prevalence rate of PTSD as a continuous measurement. Two articles assessed PTSD symptom clusters of intrusion, avoidance, and hyper-arousal. Nine of the included studies also examined PTSD rates according to a clinical cutoff point for the survey instrument.

### 3.2. Coping Measures

Among all the included studies, the Kidcope was the most commonly used measure of coping (*n* = 6) [15,17,34,35,38,40,41,42,43,44,45,46,47,48,49,50,51]. Other measures included the Simplified Coping Style Questionnaire [29,31], Coping Style Questionnaire [14,32], Children’s Coping Strategies Checklist [18,39], Brief Cope [33], Coping Strategy Scale [30], Self-Report Coping Measure (SRCM) [36], How I Coped Under Pressure Scale (HICUPS) [36], Children’s Coping Assistance Checklist [34], Rumination Response Scale [28], and Rumination Scale for Children [12]. In addition to the types of coping, two articles measured a child’s belief in their ability to cope. The measures used were the Coping Competency Beliefs [12] and Child Coping Efficacy Scale [37]. Internal consistencies for all coping measures (if reported in the original studies) can be found in Table 2.

Three different versions of the Kidcope were used in the included studies. One study [17] used an 11-item version, which appraises the frequency with which adolescents use cognitive and behavioral strategies to cope with a specific stressful event using a 4-point scale (0 = not at all to 4 = almost all of the time). This version of the Kidcope was based on a 2-factor structure of the measure tested by Cheng and Chan [49] assessing escape- and control-oriented coping styles. A 10-item version of the Kidcope was used in one study [40], separating five items into approach coping (i.e., problem-solving and cognitive restructuring) and five avoidant coping items (i.e., distraction, blaming, wishful thinking, resignation), using binary (yes/no) response options. Four studies used the 15-item version of the Kidcope [15,34,35,38], assessing how often children utilized each coping strategy (0 = not at all, 3 = all of the time); however, each study employed different scoring methods to assess coping styles. Prinstein et al. [34] constructed five two-item subscales (distraction, resignation, positive coping, people oriented and social withdrawal). Russoniello et al. [35] grouped the 15-item Kidcope to assess 10 coping constructs (distraction, social withdrawal, cognitive restructuring, self-criticism, blaming others, problem solving, emotional regulation, wishful thinking, social support, resignation). La Greca et al. [15] used a subset of three items in the 15-item version that reflected blame and anger (blaming others; yelling screaming; blaming self). Lack and Sullivan [38] combined all items to create a composite coping measure assessing the number of coping strategies participants used.

In addition to the Kidcope, Prinstein et al. [34] used the Children’s Coping Assistance Checklist (CCAC), a measure developed specifically for their study. The 30-item CCAC used a 4-point response set (0 = not at all to 3 = almost all the time) examining how often children engaged with important people in their lives (parents, teachers, and peers) for assistance in coping with trauma. Three forms of coping assistance were included in the measure: emotional processing, reinstitution of family roles and routines, and distraction. 

The Simplified Coping Style Questionnaire (SCSQ) was used in two studies. Fan et al. [29] used the SCSQ as validated by Xie [42] to measure adolescents’ use of coping styles on a 4-point scale (0 = never to 3 = often). The 20-item measure examines two constructs: positive (12 items) and negative coping (8 items). Du et al. [31] also used the SCSQ with the same items validated by Xie [42]; however, labeled the constructs active (12 items) and passive (8 items) coping.

The Coping Style Scale (CSS) was used in two studies. An et al. [14] used a subscale of the CSS to measure items related to avoidant coping. Xiao et al. [32] used the full CSS, measuring two dimensions of positive coping (problem-solving, 8 items, and resorting, 7 items) and four dimensions of negative coping (withdrawing, 5 items; abreaction, 4 items; imagining, 3 items; and tolerating, 3 items). All items in the CSS were measured along a 6-point scale (0 = never to 5 = all of the time) indicating the frequency with which participants used each coping method. 

The Children’s Coping Strategies Checklist (CCSC) was used in two studies. Pina et al. [18] used the 24-item CCSC to assess use of active and avoidant coping behaviors on a 4-point scale (1 = never to 4 = most of the time). Active coping was assessed through items examining cognitive restructuring and problem-focused strategies; avoidant coping was assessed through items exploring as repression and avoidant actions. Lengua, Long, and Meltzoff [39] used the CCSC to measure coping after the September 11th terrorist attacks, also assessing active (15 items) and avoidant (12 items) coping. 

Martin, Felton and Cole [28] used 17-items of the Rumination Response Scale (RRS) [43] which measured symptom-focused and self-focused rumination. Responses were assessed on a 4-point scale (1 = almost never, 4 = almost always) through which participants indicated how frequently they engaged in ruminative behaviors. 

Kilmer and Gil-Rivas [12] employed the “Rumination Scale for Children”, which was an adapted version of an “Adult Rumination Scale”. The 5-item scale measured how often children experience intrusive rumination and engage in deliberate rumination. 

Kilmer and Gil-Rivas [12] also used the “Coping Competency Beliefs Scale” [53], which is a five-item measure assessing children’s perception of their ability to handle problems related to a traumatic event. Higher scores indicated greater perceived coping competency. Lewis, Langley, and Jones [37] used the Child Coping Efficacy Scale, a 7-item scale to assess children’s perceived coping self-efficacy. The measure asks children to assess how successful their efforts to cope with difficult situations have been in the past, and how likely it is that they will successfully cope with future challenges. The response and scoring method for this measure was not described in the study, and the original codebook is an unpublished manuscript we were unable to access.

Chen, Wang, Zhang, and Shi [30] used the Coping Strategy Scale [44]. The 36-item measure included seven subscales: problem-solving, social support, positive cognitive restructuring, forbearance, escape, emotional expression, and wishful thinking. Chen et al. [30] grouped the seven subscales into two coping constructs, problem-focused (19 items) and emotion-focused (17 items).

Terranova [36] used two measures of coping: “The Self Report Coping Measure (SRCM)” and “How I Coped Under Pressure (HICUPS)”. Externalizing and internalizing coping subscales were used from the SRCM [42]. The externalizing coping subscale consisted of 5 items examining how often youth expressed negative emotions. The internalizing coping subscale consisted of 7-items measuring internal management of emotional reactions to stressful situations. All items assessed how often participants used each coping behavior. The authors used a modified version of the HICUPS [46] to measure avoidant coping (6 items). The authors then combined the means of the standardized means of all three subscales into a composite scale because the three dimensions are often theoretically considered to fall under a broader construct of negative coping.

The 28-item Brief Cope was used in one study. Stratta et al. [33] conducted an exploratory factor analysis (EFA) identifying three coping styles—positive, emotional, and disengagement—in a sample of Italian adolescents who completed the Brief Cope following the L’Aquila earthquake. The model identified through EFA was then used in a structural equation model to test relationships with resilience and post-traumatic stress.

### 3.3. Relationship between Coping Style and PTSD Symptoms

The included studies examined the relationships between coping styles and the severity of PTSD symptoms. Types of coping strategies that were associated with higher PTSD symptom severity included: passive, internalizing and externalizing, rumination, avoidant, negative, escape oriented, and blame and anger. Coping strategies that were inversely or weakly related to PTSD symptom severity included: active, positive, problem solving, use of social support, positive cognitive restructuring, approach, and control-oriented coping. Coping competence and coping self-efficacy were also associated with lower PTSD symptoms. Effect sizes describing the magnitude of the associations between coping style and PTSD symptoms are presented in Table 2.

Blame and anger (OR = 7.79), use of distraction (*d* = 0.78), and rumination (*r* = 0.35–0.70) appeared to have the strongest relationship with greater PTSD symptom severity. Coping efficacy (*r* = −0.42–0.47) had the greatest inverse relationship to PTSD symptom severity. Among the longitudinal studies, the relationship between coping strategies and PTSD weakened over time. For example, Terranova [36] found the association between PTSD and negative coping to reduce from (*r* = 0.47) 1.5 months after hurricane Katrina to (*r* = 0.29) 8 months post hurricane. Table 2 presents effect sizes of the relationship between coping and PTSD symptom severity.

## 4. Discussion

Post-disaster contexts are notoriously difficult settings in which to conduct rigorous research due to the many practical barriers, such as the rapid nature of response, loss of resources, damaged infrastructure, and ethical and clinical considerations for working with trauma-affected populations. Thus, researchers are often met with significant challenges in obtaining consent, recruiting participants, acquiring funding, and developing rigorous study designs [19]. Our study illustrated that accessible culturally, contextually, and developmentally validated coping measures pose as an additional barrier to conducting research in these settings. 

Of the 18 studies reviewed, two received a high quality score, 12 were deemed to be of moderate quality, and four studies were rated as fair. Strengths of the highly rated studies included measuring multiple dimensions of coping according to well-established coping theories, using psychometrically validated measures to assess coping that were both culturally and developmentally appropriate for the study samples, and explicitly accounting for challenges of the research process, such as study attrition and the presence of missing data. Features that prevented studies from receiving a rating of “high” included poor description of study practices and procedures (e.g., not describing inclusion and exclusion criteria or how missing data were handled), small sample sizes that limited the statistical power of the studies, and several issues related to the ways in which coping was measured.

### 4.1. Coping Measurement

*Measurement variance.* The primary aim of this review was to examine coping measurement in disaster-affected children and adolescents. A recurring measurement issue seen in several of the reviewed studies was the use of measures that either were not designed for research use, for which psychometrics were unavailable, or that had not been validated for the culture, situational context, or developmental age group of the participants. One example of this could be seen in the five studies that used the Kidcope. Although the Kidcope is widely used in studies of children’s coping (perhaps due to its simplicity that makes it both feasible and appealing in the chaos of post-disaster research), the measure was initially designed as an informal clinical assessment for children coping with chronic illness and was tested with a normative sample [48]. Because it was intended for use in clinical settings, psychometrics was not at the forefront of the Kidcope’s design, which is reflected in the multiple studies that have documented unstable factor structure across populations. Several authors of the reviewed studies accounted for this by running preliminary factor analysis to determine the structure of the measure in the study sample [17], referring to their own previous work with the same sample in which a factor structure was established [15], or using substantive reasoning to select a factor structure established in other work that seemed appropriate for the current study population [40]. While the authors of the reviewed studies did well with managing the psychometric instability in ways that were appropriate for their study populations, the fluctuating statistical properties of the Kidcope nonetheless limit the ability to draw valid conclusions between studies and participants of varying cultural backgrounds due to measurement variance, which in turn limits the impact of important work on children’s coping.

A less prevalent, but somewhat related measurement issue encountered in several of the studies had to do with the use of proprietary coping measures for which psychometrics could not be verified [37]. While the instruments might in fact be psychometrically strong, the fact that evidence of these properties was not referenced or established results in the same limitation as the use of the previously discussed measures. Because of potential measurement variances between groups, it is difficult to compare and contrast the results of these studies with others. The use of proprietary measures as well as measures that are not designed for research use or validated for the study populations despite their limitations may be indicative of the need for a multidimensional, comprehensive measure of children’s post-disaster coping specifically designed for use in research. 

*Development and context.* Several coping measures in the included studies were not validated for the developmental stage of participants or context of the study setting. The psychometric properties of the rumination measures (RRS and RSC), for example, were established in adult populations, yet were applied with child and adolescent participants [12,28]. The authors referenced previous research that used the measures with children and/or adolescents, however, those studies did not test the psychometric properties within younger populations [43,63]. Similarly, the SCSQ was validated with Chinese adults and college students, however, participants as young as 7th grade were included in one of the studies [29]. While scholars have advocated for the adaption of adult stress and coping measures for use with children and adolescents, the process and validation of the adapted measures should be documented [64]. Factors to consider in the adaptation include the use of child and adolescent appropriate language, context of the stressor, and developmental applicability of the coping strategy [65].

The Coping Competency Beliefs measure was validated with children of a comparable age to participants in the reviewed study; however, the psychometric properties were established with a sample of children who experienced parental divorce [53]. Because coping is influenced by both the nature and social context of an event [66], it is reasonable to assume that children might perceive their available resources and ability to successfully cope with a parental divorce differently than their capacity to cope with a sudden onset disaster. Without validation in a post-disaster context, it is unclear that assessments accurately measure children’s coping following a sudden onset disaster. Determining the usability and fit of a measure with consideration for age, demographic characteristics, and adequate psychometric properties for the study sample are all critical components to ensuring accurate interpretation of the data [67,68]. 

*Theoretical Conceptualization.* Another measurement challenge was the lack of theoretical and terminological alignment of coping. Stratta [33] conducted an exploratory factor analysis condensing 14 subscales of the Brief Cope into three coping dimensions (i.e., positive, emotional, disengagement). Disengagement coping combined subscales of humor, substance abuse, religion, and denial. Previous scholars, however, have characterized humor and religion as engagement or adaptive coping strategies [69,70]. The Kidcope grouped questions related to blame (“I blamed myself for the problem”; “I blamed others”) and anger (“I yelled screamed or got mad”) into one subscale termed “blame and anger”. While blame and anger have both been associated with higher levels of negative psychological adjustment in trauma affected children [71], the way a child internally or externally uses these strategies may result in different outcomes. For example, a child may feel angry about the disaster happening, but that anger may not result in aggressive behaviors that are associated with maladaptive outcomes. Additionally, research has consistently found associations between self-blame and psychological distress [72,73,74], however, less is known about blaming others in relation to mental health outcomes after a disaster. 

Studies that assessed positive/adaptive and negative/maladaptive strategies lacked consistent terminology. Negative, avoidant, and escape-oriented were labels used to assess strategies, such as withdrawal, avoidance, passive, internalized, externalizing behavior, and blame, which have historically been associated with higher psychological distress symptoms in trauma-affected populations [64,75]. Positive, active, approach, and control-oriented coping were terms used to group together strategies, such as problem solving, positive cognitive restructuring, and social support, which have been previously associated with lower distress symptoms [21]. 

The lack of theoretical alignment and consistent terminology limits the understanding of coping and provides continued measurement challenges. Scholars should consider establishing unified terminology and theoretical consensus to gain a more robust understanding of the effects of coping on child and adolescent well-being after a sudden onset disaster [64].

### 4.2. Self-Report Measurement

The coping measures used in the reviewed studies relied on participant self-reported information, which is appropriate and unsurprising given the personal and individualized nature of coping. Children themselves are the only respondents who are able to provide information on their internal experiences of coping, while family members and other key adults can only report on observations of children’s external behaviors and experiences. The use of self-report measures with children as the sole source of information may raise concerns about the validity and accuracy of the data obtained because of normal developmental limitations in cognitive processing and making critical inferences [76]. Because child self-report data are irreplaceable in the study of children’s coping, it is essential that children’s coping measures are developmentally validated for the target population [67]. Moving forward, measures of children’s coping should be developed with practices such as cognitive pretesting to ensure developmental validity. The use of multiple reporters would also help to mitigate developmental limitations [76]. Although parents/caregivers/teachers may not be able to identify all coping responses in young people, they can identify observable responses such as externalizing (e.g., fighting) and emotion regulation strategies which would contribute additional perspectives when assessing children and adolescent coping behaviors.

### 4.3. Coping and PTSD 

A substantive aim of this review was to examine the relationship between coping and PTSD symptoms in disaster-affected children and adolescents. All included studies examined the relationship between various coping responses and the PTSD symptom severity; however, most lacked a comprehensive assessment of coping. Many of the reviewed studies only measured dimensions of coping that were theorized to be associated with greater PTSD symptoms (e.g., rumination, avoidant) and did not include potential coping strategies that mitigate distress symptoms. Additionally, four studies did not measure specific coping strategies, but explored coping self-efficacy and/or coping assistance. Although nine studies had longitudinal designs, only three of those studies reported changes over time in the relationship between coping and PTSD.

Despite these constraints few, if any, surprises were found among the relationships between coping strategies and PTSD symptoms. In general, active or positive coping mechanisms were either inversely related to PTSD symptomology or positively correlated with symptoms at a smaller magnitude than passive or negative coping mechanisms. Youth who reported more frequent use of coping strategies, such as rumination, negative coping, emotion-focused, escape-oriented and avoidance coping, and blame and anger were more likely to experience higher PTSD symptoms than peers who did not rely as much on these coping methods. Of the three studies that reported changes over time, the association between coping and PTSD weakened over time in two studies [12,36]; and remained stable across three time points in the third study [15]. 

Belief in the ability to cope (such as reminding yourself of your capacity for resilience) and youth’s beliefs in their own capacity to cope in a healthy way were weakly correlated with PTSD symptoms [12]. Scholars have consistently found that coping self-efficacy is associated with lower rates of PTSD in trauma-affected children and adolescents [77,78]. Unfortunately, few studies in the reviewed articles incorporated measures of coping self-efficacy, and those that did, did not include additional measures examining coping styles or strategies. In post disaster settings, practitioners and researchers should assess both children and adolescents coping self-efficacy and ways of coping, which would provide a deeper understanding of most appropriate post-disaster mental health prevention and treatment interventions. 

Maintaining typical roles and routines in family and school settings was also inversely associated with PTSD symptoms [34], suggesting that consistent relationships and interactions are important to help youth manage post-disaster stress and emotions. Establishment of roles and routines is a form of coping assistance that families, clinicians, and educators can use to help children and youth restore a sense of normalcy and predictability in a post-disaster environment. 

Included studies that measured distraction identified the strategy as maladaptive/negative and related to more severe PTSD symptoms. Historically, distraction has been classified as both helpful and harmful for trauma-affected youth [6,79]. Distraction may be used to mask feelings of grief, which could potentially have a negative long-term impact on children and adolescent adjustment. Conversely, young people may use distraction to reduce rumination or continuous thoughts about the event (e.g., doing fun things to not think about the stressor). Distracting oneself as an escape- or avoidance-focused coping strategy was not especially highly associated with PTSD symptoms compared to other escape/avoidance strategies in the reviewed studies. However, having others (family, teachers, or peers) distract youth from the trauma was strongly associated with PTSD symptoms [34]. A possible explanation for why self-distraction was not as strongly associated with PTSD as distraction efforts by others may be that the distraction efforts by others are a response to signs that a young person is having trouble coping e.g., “I can see that they are struggling, so I will try to take their mind off it”—whereas youth who distract themselves might be using the strategy more effectively to manage post-disaster stress. Therefore, it is important to conceptualize how distraction is assessed to understand the relationship of this coping style with psychopathology in disaster affected children and youth.

Finally, intrusive rumination was highly correlated with PTSD symptoms [28]. This is unsurprising as research has consistently found that ruminative thoughts such as future-oriented worry and repeatedly thinking about or replaying a trauma is associated with greater PTSD symptoms [80]. This reaffirms the need for interventions focused on positive cognitive restructuring and building coping competency beliefs, which have been effective at reducing incidents of intrusive and repetitive rumination [81,82].

### 4.4. Limitations and Directions for Future Research

The purpose of the current systematic review was to evaluate measures used to assess children and adolescent coping following an acute onset disaster, and to synthesize research findings on the relationship between child/adolescent coping and PTSD. As systematic reviews are characterized by having very clear foci, we did not analyze constructs related to coping that are more suitable for a comprehensive review of the topic, such as the impact of specific coping behaviors (such as mindfulness practices) or non-disaster-related constructs such as family status (e.g., financial status, stability) or peer relationships (e.g., bullying) on coping outcomes. Additionally, we did not analyze the relationship between coping and a broader range of mental health symptoms (e.g., depression anxiety). While outside the scope of the current review, examining and synthesizing knowledge on how coping behaviors, non-disaster related constructs, and mental health symptoms such as anxiety and depression influence coping could be fruitful subjects for future reviews.

A methodological limitation to this study is that due to the information available in the published studies, we were unable to use the same effect size metric across all studies, and in one case [31], were unable to report an effect size at all. Although we could have chosen not to report effect sizes for studies that could not be converted to a common metric, we believed that the benefit of reporting as many effect sizes as possible (even in various metrics) far outweighed the alternative as far as answering our research question regarding the relationship between coping and PTSD symptoms.

It also should be noted that all studies reviewed examined coping after acute sudden onset disasters. Although there may be some aftershocks or related effects of an acute disaster after the main occurrence, for the most part acute disasters have a distinct disaster followed by a clear recovery period. For long-term disasters such as the current global COVID-19 pandemic or wars, the distinction between the active disaster and recovery periods can be blurred, overlap, or may occasionally recur. Effective mechanisms to cope with long-term disasters might be different from those that are effective in response to acute disasters, and the trajectories of coping and PTSD symptoms are also likely to be different due to the indistinct nature of disaster and recovery. Future research is needed to explore the relationship between coping and PTSD symptoms during and after long-term disasters in order to understand how it may be similar to or different from coping and PTSD symptoms in response to acute disasters.

As noted earlier, a considerable number of the reviewed studies used measures of coping that were either not validated for use with the study populations, not designed for use in research, or that did not report psychometric information. This is not necessarily the fault of the study authors, who likely used the best, most feasible tools available to them in the challenges of a post-disaster context, but indicates the need for an accessible, psychometrically rigorous measure of children and youth coping that meets the unique challenges of post-disaster research. Disaster researchers should rigorously test and establish the psychometric properties of existing youth coping measures. Scale adaptation and development work is needed to create a psychometrically sound post-disaster coping assessment and to tailor it in response to common challenges of research in post-disaster settings.

## 5. Conclusions

Understanding the relationship between coping with sudden onset disasters and post-traumatic stress in young people is critical to support their short and long-term psychological well-being. Improved knowledge of this important relationship can also guide the design and delivery of post-disaster interventions for children and adolescents. Researchers have explored the relationship between coping and PTSD symptoms; however, this review identified challenges in synthesizing research on post-disaster coping. Post-disaster environments are inherently difficult contexts in which to conduct high-quality research; therefore, brief, accessible, developmentally, and contextually appropriate coping instruments in these settings must be developed. Despite the many challenges, high-quality post-disaster research is essential to better understand how children and youth cope with and heal from the trauma of disasters [83]. 

## Figures and Tables

**Figure 1 ijerph-18-04865-f001:**
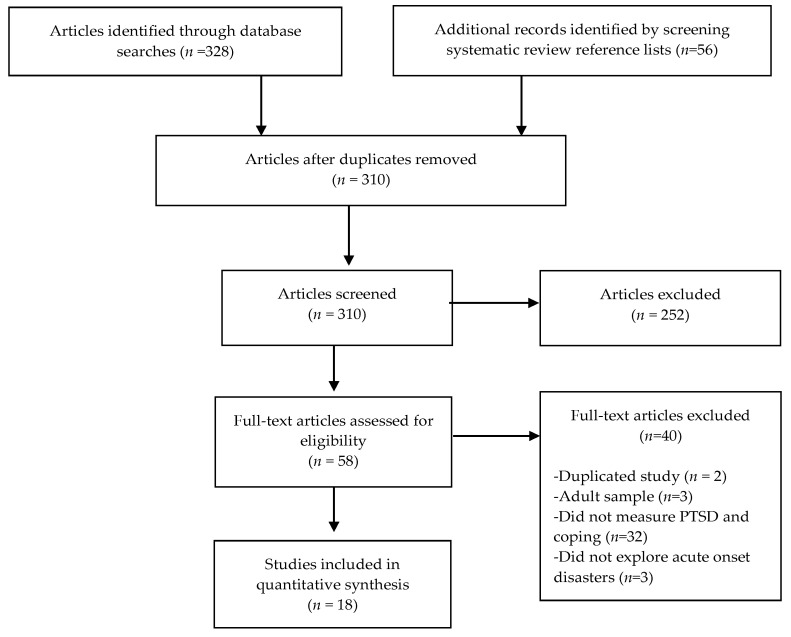
PRISMA Diagram.

**Table 1 ijerph-18-04865-t001:** Study characteristics.

Author	Event ^1^	Time Frame	Age/ Grade	*n*	Race/ Ethnicity	Country	PTSD Measure	Coping Measure	Exposure Measure	Summary of PTSD and Coping Styles ^2^	QS ^3^
**Natural Disaster**											
Martin, Felton & Cole [28]	F	Longitudinal 6-months pre flood 10-days post flood	10–15 years old	127	87% White 5% Latinx 3% Black 0.5% Asian American 6% Multiracial/ other	United States	The Child PTSD Symptom Scale (CPSS)	Rumination Response Scale (RRS)	Flood events questionnaire	Rumination (+)	Moderate
An, Fu, Wu, Lin, & Zhang [14]	EQ	Longitudinal 1-year, 1.5-years, 2-years post-EQ	13–16 years old	636	52% Quiang 26% Tibetan 18% Han 6% other	China	Child PTSD Symptom Scale (CPSS)	Coping Style Scale	N/A	Avoidant (+)	Moderate
Fan, Long, Zhou, Zheng, & Liu, [29]	EQ	Longitudinal 6, 12, 18, 24-months post-EQ	7th and 10th grade	1573	No ethnicity information included	China	Posttraumatic Stress Disorder Self-Rating Scale (PTSD-SS),	Simplified Coping Style Questionnaire (SCSQ)	4-item earthquake exposure measure	Negative (+) Positive (-)	Moderate
Chen, Wang, Zhang & Shi [30]	EQ	Cross-sectional 6-months post-EQ	4–8th grade	156	No ethnicity information included	China	Children’s Revised Impact of Event Scale (CRIES-13)	Coping Strategy Scale	7-item earthquake exposure measure	Emotion-focused (+)	Moderate
Du, Ma, Ou, Ye, Ren, & Li [31]	EQ	Cross-sectional 8-years post EQ	14–20 years	4118	99% Han 1% other	China	PTSD Checklist-Civilian Version (PCL-C)	Simplified Coping Style Questionnaire (SCSQ)	4-item earthquake exposure measure	Negative (+) Positive (-)	Moderate
Xiao, Liu, Liu, Jiang [32]	EQ	Cross-sectional 3-years post-EQ	11–18 years old	867	100% Tibetan	China	PTSD Checklist–Civilian Version. (PCL-C)	Coping Styles Scale	13-item earthquake exposure measure	Negative (+)	Moderate
Stratta et al. [33]	EQ	Cross-sectional 2-years post-EQ	17–18 years old	371	Not specified	Italy	Trauma and Loss Spectrum (TALS) Self Report	Brief Cope	N/A	Self-distraction (+) Venting (+) Denial (+) Behavioral Disengagement (+) Humor (-) Emotional support (+)	Fair
La Greca, Lai, Llabre, Silverman, Vernberg, & Prinstein [15]	H	Longitudinal 3, 7, 10-months post hurricane	3–5th grade	568	44% White 26% Latinx 22% Black 8% other	United States	PTSD Reaction Index for Children (PTSD-RI)	Kidcope 15-items	Life Events Schedule	Blame and anger (+)	Moderate
Prinstein, La Greca, Vernberg, Silverman [34]	H	Cross-sectional 7-months post hurricane	3–5th grade	506	47% White 27% Latinx 23% Black 3% Asian American	United States	PTSD Reaction Index for Children (PTSD-RI)	-Children’s Coping Assistance Checklist -KidCope 15-items	N/A	Emotional processing (+) Distraction coping assistance (+)	Moderate
Pina, Villalta, Ortiz, Gottschall, Costa and Weems [18]	H	Longitudinal 12-months pre-hurricane 6-7 months post hurricane	7–16 years old	46	67% White 33% Black	United States	The Child PTSD Checklist	Children’s Coping Strategies Checklist	13-item hurricane related experiences measure	Avoidant (+)	Fair
Kilmer, & Gil-Rivas [12]	H	Longitudinal 1- and 2-years post hurricane	7–10 years old	51	77% Black 15% White 8% other	United States	UCLA PTSD Reaction Index	-Coping Competency Beliefs -Rumination Scale for children	Hurricane-Related Exposure scale	Intrusive and deliberate rumination (+)	Moderate
Russoniello et al. [35]	H	Cross-sectional 6-months post hurricane	4th grade	150	63% Black 33% White 4% Latinx	United States	PTSD Reaction Index for Children (PTSD-RI)	Kidcope 15-items	1-item Assessing home flooding	Social withdrawal (+) Self-criticism (+) Blaming others (+) Problem solving (+)	Fair
Terranova [36]	H	Longitudinal 1–1.5 months and 2–8 months post hurricane	6th grade	175	61% White 18% Black 17% Multiple ethnicity 4% Native American/ Latinx	United States	The Child PTSD Checklist	-Self-Report Coping Measure (SRCM) -How I Coped Under Pressure Scale (HICUPS)	6-item hurricane exposure measure	Negative (+)	Moderate
Papadatou GiannopoulouBitsakou, Bellali, Talias, & Tselepi [17]	WF	Cross-sectional 6 months post wildfire	12–17 years	1468	93% Greek 6% immigrant	Greece	Children’s Revised Impact of Event Scale (CRIES)	Kidcope 11-items	Wildfire Experience Questionnaire	Escape- oriented (+)	High
Lewis Langley & Jones [37]	WF	Longitudinal 3- and 10-months post WF	14–16 years old	206	68% White 32% Black	United States	PTSD Reaction Index for Adolescents (PTSD-RI)	Child Coping Efficacy Scale	Fire-Related Traumatic Experiences	Coping efficacy (-)	Moderate
Lack & Sullivan [38]	T	Cross sectional 13-months post tornado	3–6th grade	102	90.9% White 5.5% Native American	United States	PTSD Reaction Index (PTSD-RI)	Kid Cope 15-item	Tornado exposure questionnaire	Number of coping strategies (+)	Fair
**Manmade Disaster**											
Lengua, Long, & Meltzoff, [39]	TA	Longitudinal 2–9 weeks post-9/11 terrorist attack; 6-months before 9/11 terrorist attack	9–13 years old	143	66% White 19% Black 6% Multiple ethnicity 4% Latinx 3% Asian American 2% Native American	United States	Child PTSD Symptom Scale (CPSS)	Children’s Coping Strategies Checklist	2-items assessing direct and indirect exposure to terrorist attacks	Avoidant (+)	High
**Multiple Disasters**											
Fernando & Berger [40]	TS	Cross-sectional (timeframe post TS not reported)	6–13th grade	669	77% Sri Lankan 23% Tamil 48% Buddhist 28% Muslim 15% Hindu 8% Christian	Sri Lanka	Child Post-traumatic Stress Scale (CPSS)	Kidcope Religious Coping Index	War- and Tsunami-Related Stressor Scale	Avoidant (+)	Moderate

^1.^ Event: *F* = Flood, *EQ* = Earthquake, *H* = Hurricane, *WF* = Wildfire, *Ex* = Explosion, *T* = Tornado, *TS* = Tsunami, *TA* = Terrorist Attack. ^2.^ Summary of findings PTSD and coping styles: (*+*) denotes positive relationship with PTSD symptoms; (-) inverse relationship with PTSD symptoms. ^3.^ QS: Quality Score.

**Table 2 ijerph-18-04865-t002:** Coping instrument description.

Coping Scale	Validated Populations	Response Style	Author	Positive, Active, Approach Coping Mechanisms (Internal Consistency)	Negative, Passive, Avoidant, Emotion-Focused (Internal Consistency)	Association between Coping and PTSD (Effect Size)
Simplified Coping Style Questionnaire	Chinese university students: normative population [41]	Self-report	Fan et al. [29]	Positive ɑ = 0.76	Negative ɑ = 0.65	Odds of PTSD at any wave (Fan et al.): -Medium vs. low negative coping: OR = 1.89 -High vs. low negative coping: OR = 1.73 -High vs. low positive coping: OR = 0.63
			Du et al. [31]	Active ɑ = 0.80	Passive ɑ = 0.73	N/A
Self-report coping measure	United States 4-6^th^ grade students: normative population [42]	Self-report	Terranova [36]	N/A	Negative -Internalized (inwardly managing emotional stress reactions) ɑ = 0.78 -Externalized (venting negative emotions) ɑ = 0.74	Negative coping Time 1: *r* = 0.47 Time 2: *r* = 0.29
Rumination Response Scale (RRS)	United States adults with depressive disorders [43]	Self-report	Martin Felton and Cole [28]	N/A	Rumination *Wave 1* ɑ = 0.81 *Wave 2* ɑ = 0.91	Rumination: Wave 2 *r* = 0.35
Rumination Scale for Children	Psychometric testing not conducted	Self-report	Kilmer and Gil-Rivas [12]	N/A	Rumination deliberate rumination ɑ = 0.65 Intrusive rumination ɑ = 0.33	Rumination: *r* = 0.28
Coping Strategy Scale	Chinese middle school children and adolescents: normative population [44]	Self-report	Chen, Wang Zhang, & Shi [30]	Problem Focused (problem solving, social support, positive cognitive restructuring) ɑ = 0.85	Emotion Focused (forbearance, escape, emotional expression, wishful thinking) ɑ = 0.80	**Intrusion** *Problem Focused* -Problem-solving: *r* = 0.09 -Social support: *r* = 0.01 -Positive cognitive restructuring: *r* = 0.04 *Emotion Focused* -Forbearance: *r* = 0.32 -Escape: *r* = 0.23 -Emotional expression: *r* = 0.23 -Wishful thinking: *r* = 0.22 **Avoidance** *Problem-Focused* -Problem-solving: *r* = 0.20 -Social support: *r* = 0.04 -Positive cognitive restructuring: *r* = 0.06 *Emotion-Focused* -Forbearance: *r* = 0.16 -Escape: *r* = 0.04 -Emotional expression: *r* = 0.01 -Wishful thinking: *r* = 0.08
Children’s Coping Strategies Checklist	United States 4-6th grade children: normative population [45]	Researcher administered interview	Lengua, Long, & Meltzof [39]	Active (assessing cognitive decision making, control, direct problem solving, positive cognitive restructuring, optimism, seeking understanding) ɑ = 0.90	Avoidant (assessing cognitive avoidance, avoidant actions) ɑ = 0.82	Active: *r* = 0.10 Avoidant: *r* = 0.26
			Pina et al. [18]	ɑ = 0.87	ɑ = 0.68	Active: *r* = 0.55 Avoidant: *r* = 0.58
How I coped under pressure	United States 9–13 year-old children and adolescents: normative population [46]	Self-report	Terranova [36]	N/A	Negative (avoiding situations where stressor may occur) ɑ = 0.79	Negative coping: *r* = 0.47
Coping Style Scale	Chinese middle school students [47]	Self-report	Xiao, Liu, Liu, Jian [32]	Positive (problem solving, resorting) total α = 0.88 combined positive and negative	Negative (withdrawal, abreacting, imagining, tolerating) total α = 0.88 combined positive and negative	Positive coping: OR = 0.63 Negative coping: OR = 2.85
			An et al. [14]	N/A	Avoidant (not specified) *Time 1* α = 0.77 *Time 2* α = 0.83 *Time 3* α = 0.85	Avoidant Time 1: *r* = 0.24 Time 2: *r* = 0.24 Time 3: *r* = 0.28
Kidcope 10-items	United States 10–18-year-old children and adolescents Setting: normative population and chronic illness [48]	Self-report	Fernando and Berger [40]	Approach (problem-solving and positive cognitive restructuring) α = not reported	Avoidant (distraction, blaming, wishful thinking, and resignation) α = not reported	Approach: *r* = 0.11 Avoidant: *r* = 0.16
Kidcope 11-items	Hong Kong Adolescents normative sample [49] Greek Adolescents wildfire affected sample [17]	Self-report	Papadatou et al. [17]	Control Oriented (cognitive restructuring, problem solving, social support, emotional relaxation) ɑ = 0.64	Escape Oriented (distraction, social withdrawal, self-criticism, blaming others, resignation, emotional outbursts) ɑ = 0.64	Control-oriented: Incident risk ratio (IRR) = 1.09 Escape-oriented: IRR = 1.14
Kidcope 15-items	United States 3rd–5th grade hurricane affected children [50]	Self-report	La Greca et al. [15]	N/A	Blame and Anger (blame self, blame others, yell, scream, get mad) ɑ = not reported	Odds of blame and anger coping with membership in: -Recovering trajectory of PTSD vs. resilient trajectory: OR = 5.77 -Chronic trajectory of PTSD vs. resilient trajectory: OR = 7.79 -Chronic PTSD trajectory vs. recovering trajectory: OR = 1.35
			Prinstein et al. [34]	Positive (problem solving, distraction, cognitive restructuring, social support, adaptive emotional regulation) ɑ = 0.77 Distraction ɑ = not reported	Resignation ɑ = not reported	N/A
			Russionello et al. [35]	Not designated-used individual items ɑ = 0.09–0.41		Distraction *r* = 0.02 Social Withdrawal *r* = 0.25 Cognitive Restructuring *r* = 0.07 Self-criticism *r* = 0.26 Blaming others *r* = 0.20 Problem solving *r* = 0.24 Emotional Regulation *r* = 0.40 Wishful Thinking *r* = −0.01 Social Support *r* = 0.04 Resignation *r* = 0.12
			Lack & Sullivan [38]	Combined all coping items into one composite measure ɑ = not reported		Total score *r* = 0.44
Children’s Coping Assistance Checklist	United States 3-5th grade after hurricane Andrew [34]	Self-report	Prinstein et al, [34]	Roles and Routines α = 0.78	Emotional processing ɑ = 0.74 Distraction α = 0.84	Emotional processing: *d* = 0.49 Roles and routines: *d* = −0.40 Distraction: *d* = 0.78
Brief Cope	United States adults after hurricane Andrew [51]	Self-Report	Stratta et al. [33]	Positive (planning, positive reframing, active, acceptance) α = N/A	Emotional (venting, self-blame, instrumental, support, emotional support) α = N/A Disengagement (humor, substance abuse, behavioral disengagement, denial, religion) α = N/A	Re-experiencing *Emotional:* Venting *r* = 0.30; Instrumental support *r* = 0.15; Use of emotional support = 0.22 *Disengagement:* Denial *r* = 0.31; Religion *r* = 0.07; Behavioral disengagement = *r* = 0.14 *Positive:* Humor = −0.09 *Not categorized:* Self-distraction *r* = 0.21 **Avoidance** *Emotional:* Venting *r* = 0.29 *Disengagement:* Denial *r* = 0.26; Use of emotional support *r* = 0.12 *Not categorized* Self-distraction *r* = 0.17 **Arousal** *Emotional:* Use of emotional support *r* = 0.12 *Disengagement*: Venting *r* = 0.31; Denial *r* = 0.26 *Not categorized:* Self-distraction *r* = 0.23
Child Coping Efficacy	United States 9-12 year-old children who experienced divorce [52]	Self-report	Lewis, Langley and Jones [37]	Coping Efficacy Black adolescents: ɑ = 0.86 White adolescents: ɑ = 0.85	N/A	Coping efficacy Black adolescents: *r* = −0.42 White adolescents: *r* = −0.47
Coping Competency Beliefs	United States 9–12-year-old children who experienced divorce [53]	Self-report	Kilmer & Gil-Rivas [12]	Coping Competency ɑ = 0.63	N/A	Coping competency Time 1: *r* = 0.11 Time 2: *r* = −0.20

## Data Availability

Not applicable.

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
