# Peer review of "Coping and Post-Traumatic Stress in Children and Adolescents after an Acute Onset Disaster: A Systematic Review"

_ijerph, 2021, doi:10.3390/ijerph18094865_

Round 1
Reviewer 1 Report
Thank you for doing the hard work and preparing the paper.
I would like to ask you a few questions and add a few notes if necessary.
1) With regard to coping styles and PTSD, I believe that the mindfulness perspective is also important these days. I think an addendum to this point is necessary, is it possible?
(2) In the discussion, you described the limitations of research on child psychiatry in disaster situations. There are some difficulties due to regional and cultural differences. Research on adult disaster psychiatry has the same difficulties. It was evident that simply using the adult scale was difficult.
They suggest a variety of issues. Could you consider what it would take to actually develop these coping styles and scales in different areas and developing countries?
3) Children and adolescents are especially susceptible to parental influence. There are various influences such as divorce. From this point of view, I think articles and discussion on parental coping and life stability is necessary.
(4) With regard to beliefs, there will be differences in understanding, implementation, and coping skills depending on age.
Is there any mention of whether it is necessary to separate children and adolescents in terms of age and understanding, such as by making cutoffs or drawing lines, or whether it is not necessary to separate them?
(5) Adolescents can be strongly involved in bullying, neglect, and financial problems, what about coping and this consideration?
Is there a relationship between these?
(6) Post-disaster PTSD symptoms may improve from the acute phase to the mid- to long-term phase, or they may persist chronically or worsen in a delayed manner. In addition, in children who have difficulty verbalizing, reactions may be delayed. As you can see, there are many possible cases. There are a variety of cases like this. What is your perspective on these various cases and changes over time?
Also, how about a report or discussion of how coping has been helpful from a medium to long term perspective?
Thank you for your contribution.
Author Response
We appreciate the thoughtful reviews and have addressed all feedback. Please see attachment for detailed response to the reviewers comments.

Reviewer 2 Report
Thank you for the opportunity to review the manuscript titled ‘Coping and post-traumatic stress in children and adolescents after an acute onset disaster: A systematic review’. The manuscript is well written and informative, providing a clear critique of studies conducted in the field. I commend the authors on a rigorously conducted review that represents a timely contribution to the literature. The Discussion presents a thoughtful critique of the study findings.
I would like to recommend the following minor revisions for the authors’ consideration:
- Table 2: It would be helpful if the authors included the country under ‘Validated Populations’ for each measure, even when unclear (i.e. inclusion of the original study sample, or in addition to the disaster name).
- Results: Please state the five countries represented in the sample (with %).
- Results: More detail on effective coping strategies would be useful, beyond coping efficacy.
- Discussion: Excellent opening paragraph, however the challenges in conducting research outlined by the authors are not unique to disaster-affected settings. The compressed timeframe for achieving each, within a setting affected by damaged infrastructure, loss of resources, and the ethical and clinical considerations required for working with trauma-affected populations, compounds the routine challenges in research.
- There is an absence of coping behaviours in the findings, such as alcohol and drug use, internet use, gaming, and peer-support that may be relevant to adolescents, which reflects the scope of studies included, but should be noted in the Discussion.
- The study focus is the relationship between PTSD and coping among children and adolescents, which is important, but neglects the broader range of mental health consequences of disasters. The authors could note the restricted conceptual frame within the Limitations.
Author Response

(The authors gave the same response as above.)
